# Long-Term Consequences of COVID-19 at 6 Months and Above: A Systematic Review and Meta-Analysis

**DOI:** 10.3390/ijerph19116865

**Published:** 2022-06-03

**Authors:** Yirui Ma, Jie Deng, Qiao Liu, Min Du, Min Liu, Jue Liu

**Affiliations:** 1Department of Epidemiology and Biostatistics, School of Public Health, Peking University, Xueyuan Road No. 38, Haidian District, Beijing 100191, China; 1810306205@pku.edu.cn (Y.M.); 1810306145@pku.edu.cn (J.D.); 1610306236@pku.edu.cn (Q.L.); 1510306111@pku.edu.cn (M.D.); 2Institute for Global Health and Development, Peking University, Yiheyuan Road No. 5, Haidian District, Beijing 100871, China

**Keywords:** COVID-19, long-term consequence, systematic review, meta-analysis

## Abstract

We aimed to review the data available to evaluate the long-term consequences of coronavirus disease 2019 (COVID-19) at 6 months and above. We searched relevant observational cohort studies up to 9 February 2022 in Pubmed, Embase, and Web of Science. Random-effects inverse-variance models were used to evaluate the Pooled Prevalence (PP) and its 95% confidence interval (CI) of long-term consequences. The Newcastle–Ottawa quality assessment scale was used to assess the quality of the included cohort studies. A total of 40 studies involving 10,945 cases of severe acute respiratory syndrome coronavirus-2 (SARS-CoV-2) infection were included. Of the patients, 63.87% had at least one consequence at the 6 month follow-up, which decreased to 58.89% at 12 months. The most common symptoms were fatigue or muscle weakness (PP 6–12 m = 54.21%, PP ≥ 12 m = 34.22%) and mild dyspnea (Modified Medical Research Council Dyspnea Scale, mMRC = 0, PP 6–12 m = 74.60%, PP ≥ 12 m = 80.64%). Abnormal computerized tomography (CT; PP 6–12 m = 55.68%, PP ≥ 12 m = 43.76%) and lung diffuse function impairment, i.e., a carbon monoxide diffusing capacity (DLCO) of < 80% were common (PP 6–12 m = 49.10%, PP ≥ 12 m = 31.80%). Anxiety and depression (PP 6–12 m = 33.49%, PP ≥ 12 m = 35.40%) and pain or discomfort (PP 6–12 m = 33.26%, PP ≥ 12 m = 35.31%) were the most common problems that affected patients’ quality of life. Our findings suggest a significant long-term impact on health and quality of life due to COVID-19, and as waves of ASRS-CoV-2 infections emerge, the long-term effects of COVID-19 will not only increase the difficulty of care for COVID-19 survivors and the setting of public health policy but also might lead to another public health crisis following the current pandemic, which would also increase the global long-term burden of disease.

## 1. Introduction

Caused by the severe acute respiratory syndrome coronavirus-2 (SARS-CoV-2), the pandemic of coronavirus disease 2019 (COVID-19) is currently still the greatest global public health challenge. Reported to the World Health Organization (WHO), globally, as of 5 April 2022, there have been more than 490 million confirmed cases of COVID-19, including more than 6 million deaths [1]. However, the natural history, clinical course, and long-term effects are still not fully understood [2]. While the majority of patients recover from COVID-19, for a significant number of people, the virus poses a range of serious long-term effects or complications, regardless if they are men or women, hospitalized or not, young or old, or even children [2,3]. On 6 October 2021, the WHO developed a clinical case definition of the post-COVID-19 condition by Delphi consensus: the post-COVID-19 condition occurs in individuals with a history of probable or confirmed SARS-CoV-2 infection, usually 3 months from the onset, with symptoms that last for at least 2 months and cannot be explained by an alternative diagnosis [4,5].

The post-COVID-19 condition, also known as long COVID-19, has become an important target for research and clinical practice. However, the prevalence of long COVID-19 is not the same in current studies due to the different research methods, but it seems to be significant. Statistics from Altea (a network for sharing evidence-based information on the long-term effects of COVID-19) showed that around a quarter of people who have had COVID-19 continue to experience symptoms for at least a month, and one in 10 are still unwell after 12 weeks [6,7]. Luiza’s systematic review showed that the frequency of long-term COVID-19 in the acute phase or 3–24 weeks after discharge ranged from 4.7% to 80% [8]. A meta-analysis of more than 50 long-term effects of COVID-19 showed that 80% of SARS-CoV-2 infected patients developed at least one long-term symptom, but the study defined long COVID-19 as only 14–110 days after infection with the virus, which was a little short and might overestimate the prevalence of longer-term symptoms [9].

According to the WHO, common symptoms of long COVID-19 include, but are not limited to, fatigue, shortness of breath, and cognitive impairment, and generally have an impact on daily functioning [3,4,10]. These symptoms might be new after recovery from an acute COVID-19 episode, or persistent from the initial infection [3,4,10]. Previous studies have shown that the health effects of long COVID-19 may be multi-system, including not only non-specific general symptoms but also respiratory, cardiovascular, blood, kidney, gastrointestinal, neurological, and metabolic system effects, and even thrombosis, retinal abnormalities, male erectile dysfunction, and other complications [8,11,12]. In addition, COVID-19 might be related to long-term decreased quality of life and mental health issues [13,14,15,16], a meta-analysis suggested that post-acute COVID-19 syndrome was associated with poor quality of life and persistent symptoms, including fatigue, dyspnea, anosmia, sleep disturbances, and worse mental health [15]. A recent study based on the United Kingdom Biobank (aged 51–81) reported that SARS-CoV-2 was associated with structural changes in the brain, such as changes in the frontal cortex and parahippocampal gyrus, tissue damage in areas linked to primary olfactory cortex function, and a reduction in global brain size [17]. However, it still has not been determined whether these changes were related to any long-term functional effects or whether they will return to baseline over time [18].

With the re-emergence of new waves of SARS-CoV-2 infection, long COVID-19 is expected to produce another public health crisis on the heels of the current pandemic [11]. Therefore, it is imperative to emphasize this situation and increase the awareness of medical professionals, patients, the public, and policymakers [11]. Although there have been more and more studies on the long-term effects of COVID-19, they have mostly been limited to specific systems and the conclusions were distinguished. Previous reviews have been limited to three or six months or less after the onset of acute COVID-19 and limited to specific systems, making it difficult to fully assess the longer-term effects of COVID-19. Thus, we aimed to assess the long-term effects of COVID-19 at 6 months and above to provide a more comprehensive and scientific basis for the care and rehabilitation of COVID-19 survivors, the surveillance of these patients, and setting public health policy for healthcare facilities.

## 2. Methods

### 2.1. Search Strategy and Selection Criteria

We searched studies without language restrictions in the PubMed, Embase, and Web of Science databases up to 9 February 2022 with the following search terms: (COVID-19 OR SARS-CoV-2 OR coronavirus OR long COVID-19 OR post COVID-19) AND (long-term effect OR sequelae OR consequences) AND (cohort OR follow-up OR retrospective OR prospective). We used EndNoteX8.2 (Thomson Research Soft, Stanford, CA, USA) to manage records, screen, and exclude duplicates. This study was strictly performed according to the Preferred Reporting Items for Systematic Reviews and Meta-Analyses (PRISMA). This study was registered on PROSPERO (CRD42022309720).

We included observational cohort studies that examined the long-term consequences of COVID-19 at 6 months and above. The following studies were excluded: (1) irrelevant to the subject of the meta-analysis, such as studies that did not use SARS-CoV-2 infection as the exposure; (2) insufficient data to calculate the prevalence of long-term COVID-19 consequences; (3) duplicate studies or overlapping participants; (4) reviews, editorials, conference papers, case series/reports, secondary analysis or animal experiments; (5) qualitative designs; and (6) studies that did not clarify the identification of COVID-19. For example, the confirmed diagnosis of COVID-19 via a reverse-transcription polymerase chain reaction (rt-PCR) test, serologic test, or other means was not mentioned in the text.

Studies were identified by two investigators (MYR and DJ) independently following the criteria above, while discrepancies were solved by consensus or with a third investigator (LQ).

### 2.2. Data Extraction

The following data were extracted from the selected studies: (1) basic information of the studies, including the first author, publication time, and country where the study was conducted; (2) characteristics of the study population, including the sample size, median age, gender, follow-up period, smoking status, severity of COVID-19, underlying diseases, admission to hospital or intensive care unit (ICU), and length of stay (LOS); (3) clinical features of COVID-19, including the number of cases with general COVID-19-related symptoms, respiratory symptoms, cardiovascular symptoms, gastrointestinal symptoms, and neurological symptoms, as well as the results of a pulmonary functional test (PFT) and chest computerized tomography (CT); (4) the number of cases with psychiatric problems; and (5) the number of cases with problems in 5 dimensions of the European Quality of Life Five-Dimension Five-Level Scale (EQ-5D-5L), which is an instrument developed for describing and valuing health-related quality of life by the EuroQol Group in 1987. A template was used for the primary data extraction, as shown in Appendix A.

The data extraction and determination of information eligibility were conducted by two investigators (MYR and DJ) independently following the criteria above, while discrepancies were solved by consensus or with a third investigator (LQ).

### 2.3. Quality Assessment and Risk of Bias

We used the Newcastle–Ottawa quality assessment scale to evaluate the risk of bias in the included cohort studies. Cohort studies were classified as having a low (≥7 stars), moderate (5–6 stars), or high risk of bias (≤4 stars), with an overall quality score of 9 stars. We used the Grading of Recommendations, Assessment, Development and Evaluation (GRADE) approach to evaluate the evidence quality of the long-term consequences of COVID-19. 

Quality assessment was conducted by two investigators (MYR and DJ) independently, while discrepancies were solved by consensus or with a third investigator (LQ).

### 2.4. Data Synthesis and Statistical Analysis

We performed a meta-analysis to estimate the Pooled Prevalence (PP) and its 95% confidence interval (CI) of the long-term consequences of COVID-19 at 6 months and above. We performed subgroup analyses by the follow-up period (6–12 months and ≥12 months), severity of COVID-19 (non-severe and severe; the non-severe group included mild and moderate COVID-19, and the severe group included severe and critical COVID-19), whether patients were hospitalized (inpatients and outpatients), and gender. Random-effects or fixed-effects models were used to pool the rates and adjusted estimates across studies separately, based on the heterogeneity among estimates (I²). Fixed-effects models were used if I² ≤ 50%, which represents low to moderate heterogeneity, and random-effects models were used if I² ≥ 50%, representing substantial heterogeneity. The D-L method was used to estimate the tau square in the case of random-effects models. Publication bias was assessed by Harbord’s modified test. All analyses were performed using Stata version 16.0 (Stata Corp, College Station, TX, USA).

## 3. Results

### 3.1. Basic Characteristics

In the initial literature search, 5271 potential articles were identified up to 9 February 2022 (1459 in PubMed, 1894 in Embase, 1918 in Web of Science. A total of 2512 duplicates were excluded. After reading the titles and abstracts, 2566 articles were excluded based on the inclusion and exclusion criteria. Among the 193 studies under full-text review, 153 studies were excluded. Eventually, 40 studies were included in this meta-analysis based on the inclusion criteria [19,20,21,22,23,24,25,26,27,28,29,30,31,32,33,34,35,36,37,38,39,40,41,42,43,44,45,46,47,48,49,50,51,52,53,54,55,56,57,58]. The literature retrieval flow chart is shown in Figure 1. 

The included studies were observational cohort studies describing the long-term consequences of COVID-19 at follow-up 6 months and above, which involved 10,945 cases of SARS-CoV-2 infection. A total of 26 studies described COVID-19 consequences at 6–12 months’ follow-up and 19 studies described COVID-19 consequences at 12 months and above. The majority of the included studies were of great methodological rigor (i.e., 7–9 stars on the Newcastle–Ottawa Scale); only 2 included studies had 6 stars, mainly due to the insufficient comparability between the exposed cohort and unexposed cohort. The characteristics of the included studies are shown in Appendix A.

### 3.2. Pooled Prevalence of COVID-19 Symptoms at 6 Months and Above

A total of 63.87% (95% CI, 53.64–74.09%) of COVID-19 patients reported at least one symptom at 6 to 12 months, which dropped to 58.89% (95% CI, 45.87–71.91%) at 12 months and above. COVID-19 patients are at risk for long-term symptoms from multiple systems, as shown in Table 1 and Figure 2. 

At 6 to 12 months, there were 9 symptoms with a PP of more than 20%, including mMRC = 0 (PP = 74.5%, 95% CI, 66.94–82.06%), fatigue or muscle weakness (PP = 54.21%, 95% CI, 45.16–63.27%), fatigue (PP = 30.94%, 95% CI, 20.21.41.66%), dyspnea (PP = 27.06%, 95% CI, 18.67–35.44%), anxiety (PP = 25.19%, 95% CI, 13.88–36.49%), mMRC ≥ 1 (PP = 24.49%, 95% CI, 21.17–27.81%), sleep difficulty (PP = 24.11%, 95% CI, 14.67–33.56%), difficulty concentrating (PP = 22.47%, 95% CI, 4.49–40.44%), limited mobility (PP = 21.81%, 95% CI, −4.17–47.78%), chest tightness (PP = 21.18%, 95% CI, 4.94–37.43%), and depression (PP = 20.16%, 95% CI, 10.36–29.97%). Of these nine symptoms, most were respiratory and psychiatric consequences. 

At more than 12 months’ follow-up, there were nine symptoms with a PP of more than 20%, including mMRC = 0 (PP = 80.64%, 95% CI, 62.87–98.42%), myalgia or joint pain (PP = 34.52%, 95% CI, 9.01–60.02%), fatigue (PP = 34.22%, 95% CI, 23.75–44.70%), respiratory symptoms (PP = 32.7%, 95% CI, 3.97–61.43%), rhinorrhea (PP = 30.93%, 95% CI, 11.60–50.26%), anxiety (PP = 29.78%, 95% CI, 16.29–43.27%), difficulty concentrating (PP = 29.47%, 95% CI, 19.80–39.14%), sleep difficulty (PP = 26.31%, 95% CI, 15.73–36.89%), and neurological symptoms (PP = 23.83%, 95% CI, 11.42–36.29%). Three out of the nine symptoms above were psychiatric consequences.

### 3.3. Pooled Prevalence of Pulmonary Functional Test Results after COVID-19 at 6 Months and Above

Lung function tests showed that some participants had varying degrees of reduction in lung function after COVID-19 at 6 months and above. For example, during the follow-up at 6 to 12 months, FEV1/FEV < 70% occurred in 22.86% of participants (95% CI, 8.95–36.77%). Abnormal pulmonary diffuse function (DLCO < 80%) was noteworthy, which occurred in 49.1% of the participants (95% CI, 33.27–64.9%) at 6–12 months’ follow-up and above and 31.8% (95% CI, 18.65–44.95%) at 12 months’ follow-up and above. In addition, some of this reduction seemed to taper off over time between two follow-ups; the PP of FEV1 < 80% reduced from 28.03% (95% CI, 1.04–55.03%) to 13.81% (95% CI, 5.57–22.05%), and the PP of FVC < 80% reduced from 13.66% (95% CI, 5.23–22.09%) to 12.78% (95% CI, 3.99–21.56%). The analysis results are shown in Table 1 and Figure 2.

### 3.4. Pooled Prevalence of CT Results after COVID-19 at 6 Months and Above

CT results were abnormal in 55.68% of participants at 6–12 months’ follow-up (95% CI, 26.75–84.62%), reduced to 43.76% at 12 months’ follow-up and above (95% CI, 7.78–79.74%). Fibrosis was most common at 6–12 months’ follow-up (PP = 66.28%, 95% CI, 52.35–80.21%), followed by parenchymal band (PP = 32.53%, 95% CI, 16.45–48.60%), interlobular septal thickening (PP = 21.75%, 95% CI, 6.66–36.84%), and GGO (PP = 21.25%, 95% CI, 9.79–32.71%). At 12 months’ follow-up and above, nodules were most common (PP = 38.56%, 95% CI, 18.95%–58.17%), followed by GGO (PP = 21.35%, 95% CI, 8.30–34.39%). In addition, we observed some reduction in the PP of abnormal CT results over time, such as fibrosis, which decreased significantly from 66.28% to 13.88% (95% CI, 6.04–21.72%), as well as reticular pattern and interlobular septal thickening between two follow-ups. The PP of other abnormal CT results was less than 20%. More analysis results are shown in Table 1 and Figure 2.

### 3.5. The Impact of COVID-19 on Quality of Life

Assessed by the EQ-5D-5L test, the quality of life of people with COVID-19 was affected in the long term, as shown in Table 2. Pain or discomfort and anxiety and depression were the most common, and personal care problems were the least common. At 6 to 12 months’ follow-up, 33.26% (95% CI, 27.01–39.51%) of patients had pain or discomfort problems, 33.49% (95% CI, 23.87–43.12%) had anxiety or depression problems, and only 0.94% (95% CI, 0.11–1.77%) were affected in personal care. At more than 12 months’ follow-up, 35.31% (95% CI, 22.38–48.24%) of patients had pain or discomfort problems, 35.4% (95% CI, 16.39–54.41%) had anxiety or depression problems, and 1.6% (95% CI, 0.95–2.25%) were affected in personal care.

### 3.6. Gender Differences in Consequences of Long-Term COVID

Compared to females, males with COVID-19 were less likely to develop long-term COVID-19 symptoms (OR = 0.64, 95% CI, 0.55–0.75), but only a fraction of our analysis results were statistically significant (*p* < 0.05), including fatigue (OR = 0.69, 95% CI, 0.60–0.79), headache (OR = 0.40, 95% CI, 0.25–0.65) and diarrhea (OR = 0.60, 95% CI, 0.38–0.95). Compared to females, males with COVID–19 were more likely to experience dyspnea symptoms (mMRC = 0, OR = 1.34, 95% CI, 1.01–1.77). 

In addition, female COVID-19 survivors appeared to be more likely than males to have psychological symptoms and quality of life issues. For example, there was a lower risk of anxiety (OR = 0.41, 95% CI, 0.31–0.56) and depression (OR = 0.54, 95% CI, 0.37–0.79) in male than in female COVID-19 patients. Assessed by the EQ-5D-5L, compared to female COVID-19 patients, there was a lower risk for males to experience quality of life problems, such as mobility (OR = 0.70, 95% CI, 0.51–0.91), usual activity (OR = 0.52, 95% CI, 0.31–0.85), pain or discomfort (OR = 0.74, 95% CI, 0.61–0.90), and anxiety and depression (OR = 0.55, 95% CI, 0.45–0.68). The analysis results are shown in Table 2.

### 3.7. Quality Evaluation, Risk of Bias, and Publication Bias

We evaluated the quality of all 40 included studies according to the Newcastle–Ottawa quality assessment scale, 38 of them were of good quality and had a low risk of bias (≥7 stars), and 2 were of moderate quality and moderate risk of bias (6 stars), as shown in Appendix A. We evaluated the publication bias of the included studies based on Harbord’s modified test, and the *p* values of Harbord’s modified test for all the meta-analyses were higher than 0.1, indicating that there was no publication bias.

### 3.8. GRADE Evidence Evaluation

We evaluated the evidence quality of all long-term health consequences of COVID-19 using the GRADE approach. The 40 included studies were all observational studies. After a detailed evaluation of 75 long-term COVID-19 consequences at 6–12 months’ follow-up, a total of 3 outcomes were identified as high-quality evidence, 19 outcomes were identified as moderate-quality evidence, and the remaining 53 outcomes were identified as low-quality evidence. After the assessment of 57 long-term COVID-19 consequences at 12 months’ follow-up and above, a total of 4 outcomes were identified as high-quality evidence, 17 outcomes were identified as moderate-quality evidence, and the remaining 36 outcomes were identified as low-quality evidence. The detailed results are shown in Appendix A.

## 4. Discussion

Nowadays, COVID-19 continues to ravage the world, and although the infection of the pandemic Omicron variant may be mild [29,59], that does not mean we should relax our guard. Currently, the data on the effects of COVID-19 are growing rapidly. These data suggested that even if COVID-19 patients fully recover, they may face the risk of a variety of mid- and long-term effects [60]. Our systematic review and meta-analysis of 40 cohort studies involving 10,945 cases of SARS-CoV-2 infection provide the pooled prevalence (PP) of long-term consequences of COVID-19 at 6 months and above, and we compared subgroups stratified by follow-up period, severity of COVID-19, and gender. Understanding the long-term sequelae of COVID-19 is key to early intervention, treatment, and vaccination deployment. Previous studies have looked at the COVID -19 consequences at three months or longer [61]. Our study included a longer follow-up period of 6 months or more and a more comprehensive scope, including general, cardiovascular, respiratory, gastrointestinal, and psychiatric system symptoms, as well as the evaluation of medical imaging, lung function, and quality of life.

Consistent with previous studies, the proportion of patients with at least one symptom was as high as 60% at 6 months’ follow-up, and showed a decreasing trend over time [59]. However, it should not be ignored that the proportion of patients with at least one symptom was still more than 50% when followed up at 12 months or more. In Lombardo’s study, the proportion was higher, at more than 80%, but other studies have reported a lower proportion (about 40%) [29,59]. This suggests that COVID-19 may lead to sustained effects on organs, and the inconsistent results of 12-month follow-up studies suggest that more original studies on the long-term sequelae of COVID-19 are needed. 

Available data analyses have shown that respiratory symptoms were common in long COVID-19, and a high PP of persistent dyspnea is of concern. A French study found that hyperventilation syndrome was common in COVID-19 patients (34%) [60], which may be related to the occurrence of persistent dyspnea. People with COVID-19 could suffer from varying degrees of respiratory damage. The available data showed that mild dyspnea was one of the most common symptoms in long-term COVID, and the proportions of CT abnormity and abnormal pulmonary diffuse function were reduced over time, which indicates that lung damage could be improved. In addition, we should also consider the impact of underlying respiratory conditions. In one meta-analysis, COPD patients with COVID-19 had a greater risk of severe disease than the non-COPD group (calculated Risk Ratio, RR = 1.88, 95% CI, 1.4–2.4) [61]. Another study found that COPD was associated with persistent symptoms at 12 months and above (OR = 10.74, *p* < 0.05) [59]. For people with such underlying diseases, COVID-19 sequelae may increase their burden. 

The PPs of diffuse lung function impairment (DLCO < 80%) and pulmonary fibrosis were higher at long-term follow-up, but it is encouraging that this lung damage caused by COVID-19 did not appear to develop over time. In this study, diffuse lung function impairment decreased from 50% at 6–12 months to 30% at 12 months at least, and pulmonary fibrosis decreased from 66% to 14%. A study of COVID-19 patients discharged for 12 months showed no further development of pulmonary fibrosis and progressive pulmonary interstitial changes during long-term follow-up [42]. However, it should be cautioned that the repair of pulmonary fibrosis injury may bring a great burden to patients [62].

Health-related quality of life (HRQoL) is an important indicator to evaluate the impact of diseases on patients’ physical, psychological, and social fields [63], and the EQ-5D-5L questionnaire is one of the most commonly used tools [64]. Our results suggest that COVID-19 patients may have long-term problems with quality of life and mental well-being, and that women are more likely to be affected than men. This could be because women, more than men, tend to take care of the family and the housework, and the job and income loss have caused women to face an economic crisis at the same time, as well as facing a larger burden of unpaid care [65]. In addition, women’s exposure to domestic violence has increased because of social restrictions and isolation [66].

Since the long-term effects of COVID-19 are still unclear, the best way to reduce the consequences is to avoid infection, for which vaccination is important. In addition, improving COVID-19 screening and diagnosis capabilities can help the detection and treatment as early as possible. We should pay more attention to women’s mental health and give them more psychological support, even interventions when necessary, since they are more likely to have psychological problems compared to men. In addition to the original research on the long-term effects of COVID-19, articles on the effects of vaccines on the consequences of COVID-19 are also needed. 

There is currently a lack of RCTs to evaluate interventions for the long-term impact of COVID-19. This study focused on the meta-analysis of the clinical features of the long-term impacts of COVID-19. Research studies on intervention for long-term effects of COVID-19 are recommended in the future to provide evidence-based medical evidence of high GRADE quality for the development of clinical guidelines.

Our study has some limitations. First, due to limited data, some COVID-19 consequences could only analyze the PP at either 6–12 months’ follow-up or at 12 months and more, not both. In addition, the heterogeneity of the PP for long-term COVID-19 effects was high, which may be related to age and gender differences.

## 5. Conclusions

Our results show that 63.87% of COVID-19 patients had at least one type of COVID-19 consequences at 6–12 months’ follow-up after recovery or discharge, and 58.89% of patients continued to suffer at 12 months’ follow-up and above. The most common symptoms were fatigue or muscle weakness (6–12 m: PP = 54.21%, ≥ 12 m: PP = 34.22%) and mild dyspnea (mMRC = 0) (6–12 m: PP = 74.60%, ≥12 m: PP = 80.64%). Anxiety and depression (6–12 m: PP = 33.49%, ≥12 m: PP = 35.40%) and pain or discomfort (6–12 m: PP = 33.26%, ≥12 m: PP = 35.31%) became the two most common problems affecting patients’ quality of life. Our findings suggest significant long-term impacts of COVID-19 on health and quality of life, and as waves of ASRS-CoV-2 infections emerge, the long-term effects of COVID-19 will not only increase the difficulty of the care for COVID-19 survivors and setting public health policy but also might lead to another public health crisis following the current pandemic, which would also increase the global long-term burden of disease. Therefore, the long-term effects of COVID-19 should not be ignored, and it is crucial to provide a more comprehensive and scientific basis for COVID-19 survivors to guide long-term care, rehabilitation, surveillance, and prevention measures, and to set public health policy for healthcare facilities.

## Figures and Tables

**Figure 1 ijerph-19-06865-f001:**
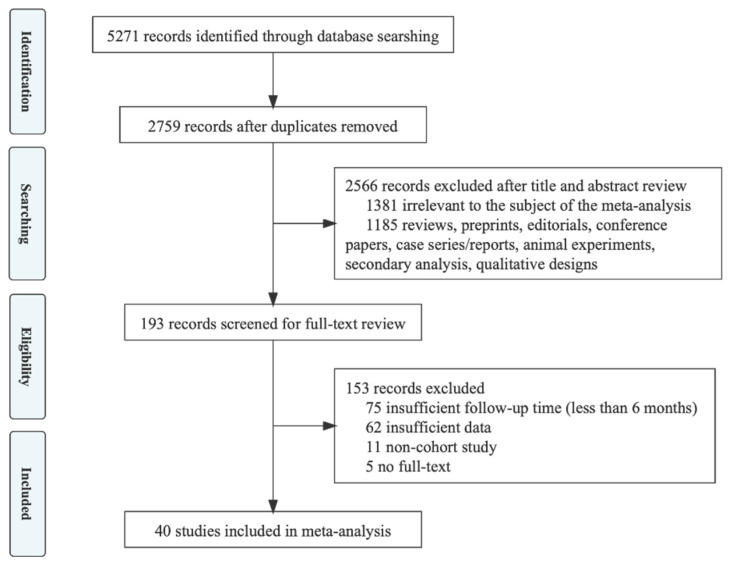
Flowchart of the study selection.

**Figure 2 ijerph-19-06865-f002:**
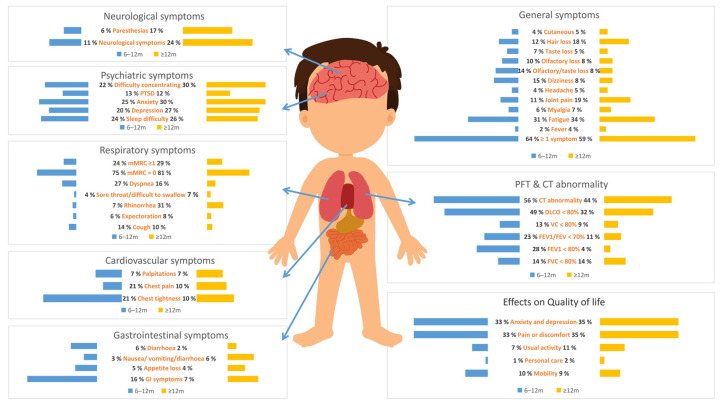
Pooled prevalence of COVID-19 consequences at 6 months and above.

**Table 1 ijerph-19-06865-t001:** Pooled prevalence of COVID-19 consequences at follow-up 6 months and above.

Consequences	6–12 Months						12 Months and Above					
	Number of Studies	Patients n/N	PP (%)	95% CI (%)	*p*-Value	I^2^	Number of Studies	Patients n/N	PP (%)	95% CI (%)	*p*-Value	I^2^
**General symptoms**												
≥1 Symptom	13	4051/6477	63.87	53.64–74.09	<0.05	98.70%	8	1230/2290	58.89	45.87–71.91	<0.05	97.20%
Fever	7	64/3403	2.07	0.32–3.82	<0.05	93.50%	7	12/778	3.53	−0.45–7.50	>0.05	70.40%
Chill	2	129/815	13.0 3	−0.33–26.39	>0.05	97.20%						
Fatigue	10	793/3000	30.94	20.21.41.66	<0.05	98.20%	14	822/3248	34.22	23.75–44.70	<0.05	98.00%
Muscle weakness	2	34/847	4.2	1.68–6.72	<0.05	15.40%	-	-	-	-	-	-
Myalgia or joint pain	-	-	-	-	-	-	2	187/503	34.52	9.01–60.02	<0.05	97.50%
Fatigue or muscle weakness	3	1949/3459	54.21	45.16–63.27	<0.05	96.40%	-	-	-	-	-	-
Limited mobility	3	83/943	21.81	−4.17–47.78	<0.05	97.90%	-	-	-	-	-	-
Myalgia	9	271/4988	6.34	3.89–8.79	<0.05	93.90%	9	128/2368	6.59	4.05–9.13	<0.05	79.80%
Joint pain	6	396/3900	11.25	7.53–14.98	<0.05	92.60%	8	320/2058	18.73	12.24 –25.22	<0.05	91.60%
Headache	8	174/5134	3.68	2.20–5.15	<0.05	89.70%	5	93/1787	5.24	3.47–7.01	<0.05	37.00%
Dizziness	5	263/3289	14.96	9.72 –20.19	<0.05	95.40%	3	100/1607	8.14	3.82–12.46	<0.05	77.70%
Olfactory or taste loss	4	161/1556	14.38	8.40–20.36	<0.05	90.20%	2	21/259	8.21	2.84–13.58	<0.05	54.60%
Olfactory loss	8	491/4507	10.07	5.47–14.68	<0.05	97.20%	8	127/2004	8.22	5.21–11.23	<0.05	70.00%
Taste loss	8	338/4507	7.48	4.46–10.50	<0.05	94.70%	7	88/2308	4.55	2.45–6.65	<0.05	78.00%
Hair loss	7	712–4485	11.58	4.08–19.08	<0.05	98.90%	4	255–1807	18.42	9.21–27.63	<0.05	94.30%
Cutaneous	7	150/4200	3.87	2.32–5.43	<0.05	84.50%	5	99/2121	4.5	3.42–5.58	<0.05	16.40%
**Respiratory symptoms**												
Respiratory symptoms	-	-	-	-	-	-	2	79/241	32.7	3.97–61.43	<0.05	96.20%
Cough	12	381/3241	13.85	9.00–18.70	<0.05	96.40%	9	81/973	9.54	5.26–13.81	<0.05	83.10%
Expectoration	3	36/554	6.45	1.01–11.90	<0.05	86.60%	4	30/488	7.97	1.23–14.71	<0.05	88.50%
Rhinorrhea	2	13/267	7.44	–5.43–20.30	>0.05	89.00%	2	68/210	30.93	11.60–50.26	<0.05	90.10%
Sore throat or difficulty swallowing	7	233/4885	4.43	2.49–6.37	<0.05	92.60%	6	78/1870	7.33	3.19–11.48	<0.05	78.20%
Dyspnea	12	717/3173	27.06	18.67–35.44	<0.05	97.60%	8	127/1129	16.43	9.66–23.20	<0.05	91.90%
mMRC = 0	5	3491/3673	74.5	66.94–82.06	<0.05	91.50%	3	1042/1448	80.64	62.87–98.42	<0.05	97.70%
mMRC ≥ 1	5	3491/3673	24.49	21.17–27.81	<0.05	76.30%	4	510/1622	29.1	10.64–47.56	<0.05	98.10%
**Cardiovascular symptoms**												
Chest tightness	2	200/815	21.18	4.94–37.43	<0.05	97.00%	3	24/278	10.24	0.77–19.71	<0.05	62.30%
Chest pain	9	265/5572	4.78	2.88–6.68	<0.05	92.20%	5	117–2009	7.76	2.60–12.91	<0.05	93.50%
Back pain	2	20/478	7.19	−3.04–17.42	>0.05	84.30%	-	-	-	-	-	-
Palpitations	5	303/3604	7.19	3.68–10.71	<0.05	93.40%	7	173/2299	6.79	3.81–9.78	< 0.05	86.00%
**Gastrointestinal symptoms**												
GI symptoms	4	87/1049	15.62	4.91–26.34	<0.05	96.80%	5	71/1178	6.57	2.48–10.65	<0.05	90.60%
Loss of appetite	7	310/5106	4.65	1.98–7.32	<0.05	96.70%	5	58/1666	3.87	1.86–5.88	<0.05	47.10%
Nausea, vomiting or diarrhea	4	126/3699	3.47	1.41–5.52	<0.05	91.80%	4	34/1550	5.86	0.73–11.00	<0.05	84.80%
Nausea	-	-	-	-	-	-	2	8/63	10.55	0.75–20.35	<0.05	41.70%
Vomiting	2	16/996	2.01	1.03–2.98	<0.05	0	-	-	-	-	-	-
Diarrhea	8	146/2272	6	2.86–9.15	<0.05	94.10%	4	8/397	2.18	–0.56–4.91	>0.05	45.90%
Stomachache	2	50/865	6.52	2.48–10.57	<0.05	40.70%	-	-	-	-	-	-
Constipation	2	22/865	6.01	−3.69–15.71	>0.05	84.50%	-	-	-	-	-	-
Altered bowel habits	-	-	-	-	-	-	2	27/165	16.17	10.56–21.78	<0.05	0.00%
**Neurological symptoms**												
Neurological symptoms	3	232/1803	10.81	0.40–21.21	<0.05	98.60%	4	167/634	23.85	11.42–36.29	<0.05	92.70%
Polyneuropathy	2	32/847	7.48	−2.95–17.91	>0.05	79.30%	-	-	-	-	-	-
Paresthesias	4	68/1305	6.24	2.24–10.24	<0.05	93.60%	4	127/679	17.42	6.90–27.95	<0.05	92.70%
Disorientation or confusion	3	37/1237	2.7	0.31–5.09	<0.05	88.10%	-	-	-	-	-	-
Forgetfulness	2	131/815	18.65	5.23–32.08	<0.05	94.80%	-	-	-	-	-	-
Memory loss	3	89/850	10.65	1.86–19.43	<0.05	96.50%	-	-	-	-	-	-
Visual impairment	3	30/760	8.11	−0.22–16.45	>0.05	89.40%	-	-	-	-	-	-
Hearing impairment	2	15/815	1.76	0.86–2.67	<0.05	0.00%	-	-	-	-	-	-
**Psychiatric symptoms**												
Sleep difficulty	9	1146/5121	24.11	14.67–33.56	<0.05	98.90%	5	476/2120	26.31	15.73–36.89	<0.05	96.20%
GAD-7 score ≥ 10	2	62/639	10.8	8.26–13.34	<0.05	-	-	-	-	-	-	-
Depression	6	301/1968	20.16	10.36–29.97	<0.05	97.30%	5	196/737	27.26	16.23–38.30	<0.05	92.30%
Anxiety	6	374/1970	25.19	13.88–36.49	<0.05	97.60%	5	213/737	29.78	16.29–43.27	<0.05	94.70%
PTSD	3	73/522	13.41	4.30–22.51	<0.05	88.70%	3	68/523	11.57	0.50–22.64	<0.05	95.70%
Difficulty concentrating	3	111/719	22.47	4.49–40.44	<0.05	96.90%	3	100/ 376	29.47	19.80–39.14	<0.05	69.50%
**PFT**												
FVC < 80%	4	45/441	13.66	5.23–22.09	<0.05	64.90%	5	43/ 374	12.78	3.99–21.56	<0.05	87.10%
FEV1 < 80%	2	11/46	28.03	1.04–55.03	<0.05	66.80%	3	28/216	13.81	5.57–22.05	<0.05	59.50%
FEV1/FEV < 70%	2	8/46	22.86	8.95–36.77	<0.05	-	3	9/ 223	4.01	−1.37–9.39	>0.05	67.10%
VC < 80%	2	43/323	13.27	9.57–16.97	<0.05	0.00%	2	22/199	11.05	6.70–15.41	<0.05	0.00%
TLC < 80%	-	-	-	-	-	-	4	28/285	9.28	4.28–14.27	<0.05	49.40%
DLCO < 80%	4	223/510	49.1	33.27–64.92	<0.05	90.60%	6	115/371	31.8	18.65–44.95	<0.05	88.10%
**CT results**												
CT abnormality	4	291/627	55.68	26.75–84.62	<0.05	98.40%	4	139/330	43.76	7.78–79.74	<0.05	98.30%
GGO	5	102/408	21.25	9.79–32.71	<0.05	89.10%	4	74/292	21.35	8.30–34.39	<0.05	87.80%
Consolidation	4	11/325	2.56	0.78–4.35	<0.05	5.50%	2	2/112	2.13	–0.79–5.04	>0.05	-
Reticular pattern	4	36/290	11.3	3.29–19.30	<0.05	80.30%	3	7/195	3.93	1.07–6.79	<0.05	0.00%
Fibrosis	3	108/272	66.28	52.35–80.21	<0.05	63.00%	3	29/209	13.88	6.04–21.72	<0.05	56.40%
Crazy paving pattern	3	0/211	-	-	-	-	-	-	-	-	-	-
Air bronchogram	3	0/211	-	-	-	-	-	-	-	-	-	-
Bronchiectasis	3	57/315	16.19	−1.82–34.21	>0.05	96.50%	2	15/180	7.39	−5.54–20.33	>0.05	91.90%
Traction bronchiectasis	2	16/93	17.63	−3.45–38.70	>0.05	86.20%	-	-	-	-	-	-
Nodules	2	20/166	9	−5.43–23.44	>0.05	92.70%	3	87/209	38.56	18.95–58.17	<0.05	87.20%
Irregular interface	2	12/93	12.88	6.07–19.68	<0.05	0.00%	-	-	-	-	-	-
Parenchymal band	2	30/93	32.53	16.45–48.60	<0.05	64.90%	-	-	-	-	-	-
Pleural effusion	3	2/284	1.69	−0.63–4.02	>0.05	-	-	-	-	-	-	-
Pericardial effusion	2	16/170	13.56	7.38–19.74	<0.05	-	-	-	-	-	-	-
Lymphadenopathy	2	5/170	4.24	0.60–7.87	<0.05	-	-	-	-	-	-	-
Interlobular septal thickening	4	63/294	21.75	6.66–36.84	<0.05	91.60%	3	14/195	7.33	1.68–12.98	<0.05	53.80%
Lines and bands	-	-	-	-	-	-	2	60/191	30.97	−0.91–62.85	>0.05	96.30%
**Quality of life evaluation (EQ-5D-5L)**												
Mobility	4	278/3421	10.36	6.88–13.83	<0.05	90.90%	2	132/1436	9.18	7.68–10.67	<0.05	0.00%
Personal care	4	40/3422	0.94	0.11–1.77	<0.05	83.20%	2	23/1436	1.6	0.95–2.25	<0.05	0.00%
Usual activity	4	129/3413	6.68	3.61–9.76	<0.05	96.30%	2	52/1436	10.76	−8.04–29.55	>0.05	97.30%
Pain or discomfort	4	989/3415	33.26	27.01–39.51	<0.05	92.60%	2	441/1436	35.31	22.38–48.24	<0.05	90.60%
Anxiety and depression	4	882/3418	33.49	23.87–43.12	<0.05	97.20%	2	406/1436	35.4	16.39–54.41	<0.05	95.60%
6MWT (distance lower than expected %)	3	1086/3258	17.33	10.64–24.02	<0.05	95.60%	-	-	-	-	-	-

Abbreviations: PP, pooled prevalence; CI, confidence interval; mMRC, Modified Medical Research Council Dyspnea Scale; GI, gastrointestinal; GAD-7, generalized anxiety disorder-7; PTSD, post-traumatic stress disorder; PFT, pulmonary functional test; FVC, forced vital capacity; FEV1, forced expiratory volume in one second; FEV1/FEV, forced expiratory volume in one second/forced expiratory volume; VC, vital capacity; TLC, total lung capacity; DLCO, carbon monoxide diffusing capacity; CT, computerized tomography; GGO, ground-glass opacity; EQ-5D-5L, European Quality of Life Five-Dimension Five-Level Scale; 6 MWT, 6-min walk test.

**Table 2 ijerph-19-06865-t002:** Gender differences in consequences of long-term COVID-19.

Consequences	Study Number	Male n/N	Female n/N	OR	95% CI	*p*-Value	I^2^
≥1 symptom	5	977/1790	1113/1749	0.64	0.55–0.75	<0.05	0.0%
**General symptoms**							
Fever	3	28/1435	32/1370	0.79	0.46–1.33	>0.05	0.0%
Fatigue	7	851/1971	942/1850	0.69	0.60–0.79	<0.05	0.0%
Muscle weakness	2	529/1284	639/1168	0.80	0.19–3.42	>0.05	92.5%
Limited mobility	3	61/619	71/577	0.76	0.51–1.15	>0.05	0.0
Myalgia	3	79/1435	93/1370	0.79	0.49–1.27	>0.05	36.9%
Headache	3	26/1435	62/1370	0.40	0.25–0.65	<0.05	0.0%
Dizziness	2	70/907	77/843	0.79	0.55–1.14	>0.05	0.0%
Olfactory or taste loss	4	223/1487	251/1413	0.85	0.69–1.04	>0.05	0.0%
Olfactory loss	2	109/1007	119/1001	0.92	0.70–1.21	>0.05	0.0%
Taste loss	2	69/1007	90/1001	0.71	0.42–1.21	>0.05	46.1%
Hair loss	2	176/1007	192/1001	0.36	0.03–3.93	>0.05	67.5%
**Respiratory symptoms**							
Cough	3	108/629	112/614	0.79	0.58–1.09	>0.05	0.0%
Sore throat or difficulty swallowing	3	68/1435	78/1370	0.79	0.57–1.11	>0.05	0.0%
Dyspnea	2	161/481	128/411	0.82	0.31–2.16	>0.05	75.6%
mMRC = 0	2	786/1023	709/987	1.34	1.01–1.77	<0.05	30.6%
mMRC ≥ 1	2	237/1023	278/955	0.64	0.36–1.11	>0.05	78.6%
**Cardiovascular symptoms**							
Chest pain	3	66/1284	62/1168	0.96	0.67–1.37	>0.05	0.0%
**Gastrointestinal symptoms**							
GI symptoms	3	101/618	105/576	0.83	0.61–1.13	>0.05	0.0%
Loss of appetite	2	73/1284	73/1168	0.92	0.66–1.30	>0.05	0.0%
Nausea, vomiting or diarrhea	3	78/1435	104/1370	0.65	0.41–1.03	>0.05	38.7%
Diarrhea	2	36/579	50/571	0.60	0.38–0.95	<0.05	0.0%
**Neurological symptoms**							
Paresthesias	2	49/566	62/534	0.99	0.35–2.76	>0.05	78.3%
**Psychiatric symptoms**							
Sleep difficulty	4	325/1474	365/1377	0.75	0.52–1.07	>0.05	52.6%
Depression	3	93/776	112/641	0.54	0.37–0.79	<0.05	21.3%
Anxiety	3	104/775	152/640	0.41	0.31–0.56	<0.05	0.0%
**Quality of life evaluation (EQ-5D-5L)**							
Mobility	2	69/1046	92/1005	0.70	0.51–0.91	<0.05	0.0%
Personal care	2	6/1047	7/1005	0.82	0.27–2.45	>0.05	0.0%
Usual activity	2	25/1041	45/1000	0.52	0.31–0.85	<0.05	0.0%
Pain or discomfort	2	266/1044	316/1000	0.74	0.61–0.90	<0.05	0.0%
Anxiety and depression	2	200/1046	300/1001	0.55	0.45–0.68	<0.05	0.0%

Abbreviations: CI, Confidence interval; OR, odds ratio; mMRC, Modified Medical Research Council Dyspnea Scale; GI, gastrointestinal; EQ-5D-5L, European Quality of Life Five-Dimension Five-Level Scale.

## Data Availability

Data are available from the corresponding author by request.

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
