# Peer review of "Long-Term Consequences of COVID-19 at 6 Months and Above: A Systematic Review and Meta-Analysis"

_ijerph, 2022, doi:10.3390/ijerph19116865_

Round 1

Reviewer 1 Report

The authors have performed a meta- analysis of the patients recovered from Covid- 19 at 6 and post 6 months. Approximately 63.87 % and 58.89% of the recovered patients had at least one symptom after 6 and 12 months of recovery. The most common symptom being fatigue and breathing problem. Lung function impairment, abnormal CT, depression, anxiety also effected the patient’s life. This article has raised the concern for post recovery monitoring of recovered patients to avert a public health crisis.  

The manuscript has been drafted well. The material- method section and results section is clear to understand.

The authors have done a commendable job collecting and presenting the data. It is quite relevant to the after- effects of the current pandemic. I just have a minor correction which I have mentioned below.

  • Line 55, 311: SARS is written as ASRS.

Author Response

Line 55, 311: SARS is written as ASRS.

Response:Thanks for the reviewer’s suggestion. Sorry for the typo. We have corrected the spelling from ASRS to SARS.

Reviewer 2 Report

The authors have addressed an important point through the systematic review to look at the long term consequences of COVID-19 at 6 and 12 months. The review adopted rigorous methodological standards. It still can be improved by stating the outcomes clearly and including GRADE assessment as meta-analysis is performed on the data.
Abstract
1. Methodology is not included. Briefly indicate the search, quality assessment etc.
Introduction
2. The authors stated in their aim that they aim to understand the long term consequences of COVID which is a very broad aim. What are the outcomes they are looking at? You need to state the specific outcomes (primary/secondary) that you will be using to achieve the aim.
3. What is PP and mMRC? Please expand the abbreviations. It makes very difficult to read the article with so many abbreviations without expansions given.
Methdology
3. Although it is stated in L96 that the review strictly followed PRISMA reporting, the reporting format at certain sections doesn’t align with PRISMA.
e.g. abstract: PRISMA requires reporting of data source, eligibility etc in the abstract. 
4. L111: what is public time? Is it publication year?
5. L112: study country? Country where the study was conducted?
6. Was the data extraction template piloted? How did you handle articles with unclear information, eg. Where eligibility couldn’t be determined? 
7. How did you handle secondary analysis?
8. was case series /reports included?
9. Was qualitative designs included, if not why?
10. Since this review is on an important topic the quality of the evidence needs to be graded using GRADE criteria. Since the heterogeneity is also substantial this is an essential step. That will help the readers, especially clinicians to make use of the results for clinical decisions.
Results
11. L148 says the db used was web of science and L89 says science direct. Which database was used?
12. Figure 1: what type of study designs were excluded?
13.  L157-158: 26 studies described 6-12 months and 19 studies above 12 months. But you have only 40 included studies. Was there an overlap where studies split data at 2 follow up period?
14. The layout of table 1 is makes the data difficult to read. Transfer the table in landscape orientation.
15. section 3.2. what is PP? try to avoid abbreviations in title
16.” 63.87% (95%CI, 53.64-74.09) of COVID-19 patients reported at least one symptom at 
6 to 12 months, dropping to 58.89” The meaning is unclear. 
17. “COVID-19 patients are at risk for mid-and-long term symptoms. “. Mid and long term symptoms? 
18. Please use foot notes for the tables to explain the abbreviations.
19. section 3.5: QoL was explained in the context of comorbidities. Was there a scale that measured the QoL and if the scores of that scales were used to get a reliable score?
20. Consequences of male and female COVID-19 patients. Please rephrase. consider “gender differences in the consequences of long term covid “. Anything that is grammatically correct.
21. Compared with females…more likely to occur… compared to female COVID 19 patients. more likely to show. There are a lot of grammatical mistakes. Please let a native speaker proof read the article. I am skipping those errors for the rest of the manuscript.
21. L241: there were no publication bias
22. Please include a section of RoB and quality of the evidence (GRADE).

Discussion
23. PP (pooled prevalence) is explained for first time in the discussion while the abbreviation was used in the abstract.
The discussion is too concise and doesn’t reflect the clinical implications of long term effects. What are the clinical implications of the individual consequences? Systematic reviews are meant to lead way to clinical decisions. Without this information the review will not be helpful. Also the information on the quality of evidence needs to be included and discussed. The authors indicate that covid -19 symptoms subside substantially by 12 months. Does that mean 6 months’ period is the critical period and do you/or the included studies give any recommendations on the management of these patients?

Author Response

Abstract
1. Methodology is not included. Briefly indicate the search, quality assessment etc.

Response: Thanks for the reviewer’s suggestion. We have review the manuscript and added methodology to Abstract.

Introduction
2. The authors stated in their aim that they aim to understand the long term consequences of COVID which is a very broad aim. What are the outcomes they are looking at? You need to state the specific outcomes (primary/secondary) that you will be using to achieve the aim.

Response: Thanks for the reviewer’s suggestion. We are looking at the pooled prevalence of clinical symptoms in different systems (including respiratory and extrapulmonary symptoms), CT results, pulmonary function test, and quality of life after infection of SARS-CoV-2. We have revised it in the part of Data extraction.

3. What is PP and mMRC? Please expand the abbreviations. It makes very difficult to read the article with so many abbreviations without expansions given.

Response: Thanks for the reviewer’s suggestion. PP refers to the pooled prevalence, mMRC refers to Modified Medical Research Council Dyspnea Scale. We have explained the abbreviations in the foot notes below each table and abbreviation list behind conclusion.

3. Although it is stated in L96 that the review strictly followed PRISMA reporting, the reporting format at certain sections doesn’t align with PRISMA.
e.g. abstract: PRISMA requires reporting of data source, eligibility etc in the abstract. 

Response: Thanks for the reviewer’s suggestion. We have reviewed and added these sections to our manuscript.

4. L111: what is public time?Is it publication year?

Response: Thanks for the reviewer’s suggestion. It ‘s publication time, we have revised it.

5. L112: study country? Country where the study was conducted? 

Response: Thanks for the reviewer’s suggestion. It’s country where the study was conducted, we have revised it.

6. Was the data extraction template piloted? How did you handle articles with unclear information, eg. Where eligibility couldn’t be determined? 

Response: Thanks for the reviewer’s suggestion. Data extraction was conducted by two investigators independently. If they couldn’t determine the eligibility of study, it would be solved by the third investigator.

7. How did you handle secondary analysis?

Response: Thanks for the reviewer’s suggestion. We excluded studies with secondary analysis.

8. was case series /reports included?

Response: Thanks for the reviewer’s suggestion. We excluded case series /reports.

9. Was qualitative designs included, if not why?

Response: Thanks for the reviewer’s suggestion. We didn’t included qualitative designs. Because we aim to calculating prevalence of outcomes to evaluate the long-term covid and we need more objective data. The data collected in qualitative research are greatly influenced by the subjects' subjectivity.

10. Since this review is on an important topic the quality of the evidence needs to be graded using GRADE criteria. Since the heterogeneity is also substantial this is an essential step. That will help the readers, especially clinicians to make use of the results for clinical decisions.

Response: Thanks for the reviewer’s suggestion. GRADE criteria are mainly used to evaluate the level of evidence for each recommended item when producing clinical guidelines, usually for the evaluation of interventions, and rarely for single meta-analysis. We agree with the reviewer's opinion, and we also believe that the results of this study are of great significance for clinicians' practice. Therefore, referring to the suggestions of reviewers, we added the content about Grade grading in the discussion.

Results
11. L148 says the db used was web of science and L89 says science direct. Which database was used?

Response: Thanks for the reviewer’s suggestion. We used web of science instead of science direct, and we have modified the expression of L89.

12. Figure 1: what type of study designs were excluded?

Response: Thanks for the reviewer’s suggestion. We excluded studies other than cohort studies, and the inclusion and exclusion criteria were specified in the methodology.

13. L157-158: 26 studies described 6-12 months and 19 studies above 12 months. But you have only 40 included studies. Was there an overlap where studies split data at 2 follow up period?

Response:Thanks for the reviewer’s suggestion. Some of the included studies analyzed long-term health effects of COVID-19 at both 6-12 months and above 12 months follow-up, which we performed separately after a rigorous assessment of feasibility, with no overlap between the two.

14. The layout of table 1 is makes the data difficult to read. Transfer the table in landscape orientation.

Response:Thanks for the reviewer’s suggestion. Due to the requirements of the magazine layout style, we are afraid that we cannot Transfer the table in landscape orientation, so we will try to upload the “Table 1. Characteristic of the included studies.” as an supplementary table.

15. section 3.2. what is PP? try to avoid abbreviations in title

Response:Thanks for the reviewer’s suggestion. PP means pooled prevalence and we have revised the abbreviation in title.

16.” 63.87% (95%CI, 53.64-74.09) of COVID-19 patients reported at least one symptom at 
6 to 12 months, dropping to 58.89” The meaning is unclear. 

Response: Thanks for the reviewer’s suggestion. We have revised it: 63.87% (95%CI, 53.64%-74.09%) of COVID-19 patients reported at least one symptom at 6 to 12 months, dropping to 58.89% (95%CI, 45.87%-71.91%) at 12 months and above.

17. “COVID-19 patients are at risk for mid-and-long term symptoms. “. Mid and long term symptoms? 

Response: Thanks for the reviewer’s suggestion. We have corrected “mid-and-long term symptoms” to “long-term symptoms”.

18. Please use foot notes for the tables to explain the abbreviations.

Response: Thanks for the reviewer’s suggestion. We have added the foot notes below each table to explain the abbreviations.

19. section 3.5: QoL was explained in the context of comorbidities. Was there a scale that measured the QoL and if the scores of that scales were used to get a reliable score?

Response: Thanks for the reviewer’s suggestion. Health-related quality of life (HRQoL) is an important indicator to evaluate the impact of diseases on patients' physical, psychological and social fields. The QoL data we extracted were measured by EQ-5D-5L questionnaire, which is one of the most commonly used in the world and has good reliability and validity.

20. Consequences of male and female COVID-19 patients. Please rephrase. consider “gender differences in the consequences of long term covid “. Anything that is grammatically correct.

Response: Thanks for the reviewer’s suggestion. We have revised it: “3.6. Gender differences in consequences of long-term coivd”.

21. Compared with females…more likely to occur… compared to female COVID 19 patients. more likely to show. There are a lot of grammatical mistakes. Please let a native speaker proof read the article. I am skipping those errors for the rest of the manuscript.

Response: Thanks for the reviewer’s suggestion. We have reviewed the manuscript and revised the sentence.

21. L241: there were no publication bias

Response: Thanks for the reviewer’s suggestion. We have revised the sentence.

22. Please include a section of RoB and quality of the evidence (GRADE).

Response: Thanks for the reviewer’s suggestion. We also believe that Rob, level of evidence and quality could help clinicians make clinical decisions and guide patient rehabilitation. According to some previous studies such as “Scurt FG, Ewert L, Mertens PR, Haller H, Schmidt BMW, Chatzikyrkou C. Clinical outcomes after ABO-incompatible renal transplantation: a systematic review and meta-analysis. Lancet. 2019;393(10185):2059-2072. doi:10.1016/S0140-6736(18)32091-9”, “Liu Q, Qin C, Liu M, Liu J. Effectiveness and safety of SARS-CoV-2 vaccine in real-world studies: a systematic review and meta-analysis. Infect Dis Poverty. 2021;10(1):132. Published 2021 Nov 14. doi:10.1186/s40249-021-00915-3”, the Newcastle-Ottawa quality assessment scale (NOS) was used most commonly to evaluate the literature quality of observational cohort studies. Based on the advice of the reviewer, we have listed the results of NOS to evaluate the risk of bias independently and uploaded as supplementary table 2. As for GRADE criteria, it is mainly used to evaluate the level of evidence for each recommended item in the guidelines in the formulation of clinical guidelines, usually for the evaluation of interventions, and rarely for single meta-analysis. We agree with the reviewer's advice, and we also believe that the results of our study are of great significance for clinicians' practice. Therefore, referring to the advice of the reviewer, we added the content about Grade criteria in the section of discussion.

Discussion 
23. PP (pooled prevalence) is explained for first time in the discussion while the abbreviation was used in the abstract.
The discussion is too concise and doesn’t reflect the clinical implications of long term effects. What are the clinical implications of the individual consequences? Systematic reviews are meant to lead way to clinical decisions. Without this information the review will not be helpful. Also the information on the quality of evidence needs to be included and discussed. The authors indicate that covid -19 symptoms subside substantially by 12 months. Does that mean 6 months’ period is the critical period and do you/or the included studies give any recommendations on the management of these patients?

Response: Thanks for the reviewer’s suggestion. We have reviewed the manuscript and revised the discussion. People with COVID-19 could suffer from varying degrees of respiratory damage. Available data showed mild dyspnea was one of most common symptom in long term covid, the proportion of CT abnormity and abnormal pulmonary diffuse function reduce by time, which indicate the lung damage could be improved. Besides, we should strengthen the protection of people with underlying conditions like COPD, since they had greater risk of severe disease according to previous study. We should also pay attention to women's mental health and give them more psychological support since they were more likely to have psychological problems. We are not sure if 6 months’ period is the critical period for the treatment and surveillance or not since the long-term effects of COVID-19 is still unclear, more original researches are needed.

Round 2

Reviewer 2 Report

The authors have addressed majority of the queries. However, for some queries the changes are not made in the mansucript. For systematic review it is very important to describe the data collection method clearly. Eg. Query #6: It is a requirement that authors should indicate if the data collection form had been piloted (my query was not answered if yes/ no), if not the reason. Piloting data collection form is not the usual process of data collection but selection of n number of article and test if the template is good enough to capture all relevant data and if the reviewers are capturing the data in a similar way. It will help to identify any changes in the definition of fields and addition of extra fields. Also, it is unlikely that all articles will have necessary information regarding their methodology that will help reviewers to decide on their eligibility. In such cases we contact the authors for clarification and decide afterwards. Thus your methodology must include the details of your responses for Query 6-9, which is not reflected. Again I disagree on your point on GRADE. It is alright to exclude GRADING (which will impact the article) as long a s you have a valid explanation. GRADE is done for all major outcomes in reviews with even just observational studies. So I leave it to the authors to give a valid explanation on why GRADE is not included.

Author Response

(1) For systematic review it is very important to describe the data collection method clearly. Eg. Query #6: It is a requirement that authors should indicate if the data collection form had been piloted (my query was not answered if yes/ no), if not, the reason. Piloting data collection form is not the usual process of data collection but selection of n number of article and test if the template is good enough to capture all relevant data and if the reviewers are capturing the data in a similar way. It will help to identify any changes in the definition of fields and addition of extra fields. Also, it is unlikely that all articles will have necessary information regarding their methodology that will help reviewers to decide on their eligibility. In such cases we contact the authors for clarification and decide afterwards. Thus your methodology must include the details of your responses for Query 6-9, which is not reflected.

Response: Thanks for the reviewer’s suggestion. We really appreciate the recommendations of your query 6-9. As for query 6, the data collection form was piloted. We have added the description in the revised version. In addition, we added a data collection form in the supplemental table. Please refer to the methods and supplemental files. We have added the details of our responses for Query 6-9 in the revised manuscript. Thanks again for the valuable comments and suggestions.

(2) Again I disagree on your point on GRADE. It is alright to exclude GRADING (which will impact the article) as long a s you have a valid explanation. GRADE is done for all major outcomes in reviews with even just observational studies. So I leave it to the authors to give a valid explanation on why GRADE is not included.

Response: Thanks for the reviewer' s suggestion. According to the suggestion of reviewer, we revised the methodology, included the results of GRADE evidence evaluation in section 3.8, and uploaded it as a supplementary file.